# Belief-Dependent Macro-Action Discovery in POMDPs using the Value of Information

**Genevieve Flaspohler**[1,2]**, Nicholas Roy**[1]**, and John W. Fisher III**[1]
Massachusetts Intitute of Technology[1] and the Woods Hole Oceanographic Institution[2]
{geflaspo, nickroy, fisher}@csail.mit.edu

## Abstract

This work introduces macro-action discovery using value-of-information (VoI) for robust and efficient planning in partially observable Markov decision processes (POMDPs). POMDPs are a powerful framework for planning under uncertainty. Previous approaches have used high-level *macro-actions* within POMDP policies to reduce planning complexity. However, macro-action design is often heuristic and rarely comes with performance guarantees. Here, we present a method for extracting belief-dependent, variable-length macro-actions directly from a low-level POMDP model. We construct macro-actions by chaining sequences of open-loop actions together when the task-specific value of information (VoI) — the change in expected task performance caused by observations in the current planning iteration — is low. Importantly, we provide performance guarantees on the resulting VoI macro-action policies in the form of bounded regret relative to the optimal policy. In simulated tracking experiments, we achieve higher reward than both closed-loop and hand-coded macro-action baselines, selectively using VoI macro-actions to reduce planning complexity while maintaining near-optimal task performance.

## 1 Introduction

Partially observable Markov decision processes (POMDPs) are a powerful and general framework for model-based planning under uncertainty [9]. A core challenge in POMDP planning is that the optimally reachable belief space $\mathcal{R}^*(b_0)$ — the set of beliefs that are reachable from an initial belief $b_0$ under stochastic observation transitions when following an optimal policy — grows exponentially with the planning horizon in the size of the observation set. The complexity of computing an optimal POMDP policy is related to the covering number of $\mathcal{R}^*(b_0)$ [13], and this exponential growth poses a challenge for planning algorithms that attempt to approximate $\mathcal{R}^*(b_0)$ using offline, point-based approximations [11, 15, 17] or online, sampling methods [18–20].

Previous approaches have introduced high-level macro-actions [1] or options [21], such as *drive to the nearest exit*, to reduce planning complexity in complex tasks. Policies that use open-loop macro-actions have the dual benefits of a shorter effective planning horizon and smaller reachable belief space (RBS), as a policy's reachable belief space grows linearly rather than exponentially when acting in open-loop (Figure 1). However, macro-actions are largely hand-coded [2, 6, 22] or learned without formal guarantees [1, 3, 8]. Here, we address the key challenge of generating macro-actions from a low-level POMDP model such that the resulting policies have bounded regret.

This paper introduces a method for generating belief-dependent, variable-length macro-actions using a point-based representation of the POMDP value function. Our key insight is to introduce a value of information (VoI) function — which estimates the change in expected task performance caused by sensing in the current planning iteration — and constrain policies to selectively act open-loop

when VoI is low. Unlike hand-coded or learned macro-actions, we show that a horizon-$H$ policy utilizing VoI macro-actions has bounded regret $r_H$ compared to the optimal policy. Letting $V_H^*$ be the expected reward of an optimal policy and $V_H^{MA}$ the expected reward of the VoI macro-action policy, our main result (Theorem 5.2) shows:

$$r_H = \left\| V_H^* - V_H^{MA} \right\|_\infty \leq \frac{1 - \gamma^H}{1 - \gamma} \left( \delta_{\mathcal{B}} \left( 3L + \frac{R_{max}}{1 - k\gamma} + L\gamma k \right) + \tau \right), \tag{1}$$

where $\gamma$ is the POMDP discount factor, $L$ is a Lipschitz constant describing the smoothness of the value function in belief space, and the POMDP reward function is bounded in $[-R_{max}, R_{max}]$.

The three remaining terms — $\tau$, $\delta_{\mathcal{B}}$, and $k$ — elucidate the key trade-offs for macro-action-based POMDP planning. Introducing potentially sub-optimal macro-actions into a policy increases regret. The parameter $\tau$ is a VoI threshold, below which the planner acts in open-loop; high values of $\tau$ increase macro-action utilization but also increase regret. However, macro-action policies are often easier to approximate than an optimal policy. During planning, we approximate the value function at a set of beliefs that form a $\delta_{\mathcal{B}}$-covering of the macro-action policy's RBS. Since the open-loop belief dynamics are a $k$-contractive mapping on belief space, this RBS grows slowly when acting in open-loop; macro-action utilization leads to lower values of $\delta_{\mathcal{B}}$ and lower regret bounds. The form of Eq. 1 makes the trade-off between policy complexity, as measured by the size of a policy's reachable belief space, and policy performance explicit. Somewhat surprisingly, although consistent with Eq. 1, our empirical results demonstrate that macro-action policies can even outperform approximations of the optimal policy when planning with a finite point-based belief representation.

In the following sections, we introduce VoI macro-action generation and present empirical results in a set of simulated tracking experiments. Taken together, VoI macro-action generation and the associated regret bound address two fundamental questions for macro-action-based planning in partially observable domains: how do we construct high-value macro-actions and when can we use them without compromising policy performance?

## 2 Related Work

Existing offline [11, 15, 17] and online [18, 19] POMDP planners must contend with the rapid growth of $\mathcal{R}^*(b_0)$ and the resulting difficulty of approximating optimal plans. Previous work has quantified the hardness of approximating optimal POMDP policies in terms of the covering number of $\mathcal{R}^*(b_0)$ [13] and POMDP planners such as SARSOP [11] leverage this insight during planning. Online POMDP solvers, on the other hand, search over a reduced RBS by sampling scenarios in a receding horizon fashion [18–20]. However, the performance of many online planning algorithms depends on the complexity of the optimal policy [19]. For problems in which the covering number of $\mathcal{R}^*(b_0)$ is large, both offline and online methods have little recourse. By contrast, we explicitly search for near-optimal policies that are easy to approximate by selectively employing open-loop macro-actions to reduce the size of the policy's reachable belief space.

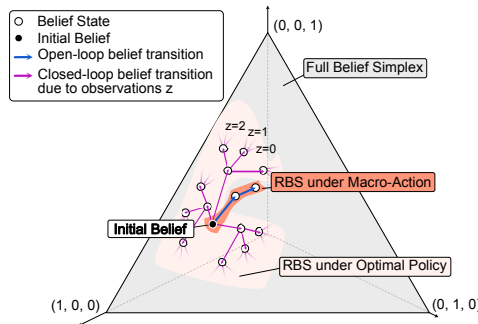

*Figure 1: **Reachable Belief Space (RBS) and Macro-actions:** POMDP planning algorithms often reason over the value of beliefs in a policy's reachable belief space (RBS). However, the size of a policy's RBS generally grows exponentially with the planning horizon in the size of the observation set $\mathcal{Z}$. This exponential growth is visualized for a three-state discrete POMDP with $|\mathcal{Z}| = 3$. Because the belief transitions deterministically under the open-loop VoI macro-actions, the size of the RBS grows only linearly during macro-action execution.*

Options and macro-actions [21] have been widely used within the POMDP and reinforcement learning communities to reduce planning complexity. Previous approaches use a presribed set of macro-actions or closed-loop options, which are identified to provide a useful problem decomposition [2, 6, 7, 10, 14, 22]. These algorithms allow planners to search over shorter effective planning horizons and often benefit from a reduced RBS, but do not provide a mechanism to identify useful options or macro-actions from the underlying planning problem. Recently, work in deep reinforcement learning has attempted to directly learn data-dependent closed-loop options in fully-observeable problems [1, 3, 4, 8]. However, these approaches do not provide formal performance guarantees or deal with the growth of the RBS and the other challenges present in partially observable problems.

## 3  Planning Preliminaries

POMDPs are general framework for planning under uncertainty. Let $\Pi(\cdot)$ denote the space of probability distributions over the argument. A finite-horizon POMDP can be represented as tuple: $(\mathcal{S}, \mathcal{A}, T, R, \mathcal{Z}, O, b_0, H, \gamma)$, where $\mathcal{S}$ are the states, $\mathcal{A}$ are the actions, and $\mathcal{Z}$ are the observations. At planning iteration $t$, the agent selects an action $a \in \mathcal{A}$ and the transition function $T : \mathcal{S} \times \mathcal{A} \to \Pi(\mathcal{S})$ defines the probability of transitioning between states in the world, given the current state $s$ and control action $a$. After the state transition, the agent receives an observation according to the observation function $O : \mathcal{S} \times \mathcal{A} \to \Pi(\mathcal{Z})$, which defines the probability of receiving an observation, given the current state $s$ and previous control action $a$. The reward function $R : \mathcal{S} \times \mathcal{A} \to \mathbb{R}$ serves as a specification of the task. A POMDP is initialized with belief $b_0$ and plans over horizon $H$ with discount factor $\gamma$. In the following, we consider finite-horizon planning problems; extensions of many of the results to discounted infinite-horizon problems is straightforward.

Due to the stochastic and partially observable nature of current and future states, the realized reward in a POMDP is a random variable. Optimal planning is often defined as finding the sequence of policies $\{\pi_t^* : \Pi(\mathcal{S}) \to \mathcal{A}\}_{t=0}^{H-1}$ that maximize expected reward: $\mathbb{E}\Big[ \sum_{t=0}^{H-1} \gamma^t R\big(S_t, \pi_t(b_t)\big) \mid b_0 \Big]$, where $b_t$ is the updated belief at time $t$, conditioned on the history of actions and observations.

The recursively defined horizon-$h$ optimal value function $V_h^*$ quantifies, for any belief $b$, the expected cumulative reward over the remaining planning iterations when following an optimal policy: $V_0^*(b) = \max_{a \in \mathcal{A}} \mathbb{E}_{s \sim b}[R(s,a)]$ and

$$V_h^*(b) = \max_{a \in \mathcal{A}} \mathbb{E}_{s \sim b}[R(s,a)] + \gamma \int_{\mathcal{Z}} P(z \mid b,a) V_{h-1}^*(b^{a,z}) \mathrm{d}z \qquad h = 1, \dots, H-1, \qquad (2)$$

where $b^{a,z}$ is the updated belief after taking control action $a$ and receiving observation $z$, computed via Bayes rule using the transition $T$ and observation $O$ functions. The optimal policy at horizon $h$ is to act greedily according to a one-step look ahead of the value function.

## 4  Generating Belief-Dependent Macro-Actions

In the following section, we introduce value of information (VoI) and describe how VoI can be used to generate belief-dependent, variable-length macro-actions. In Eq. (4), we define VoI for a given belief as the change in expected long-term reward caused by acting closed-loop and collecting a sensor observation in the current planning iteration. Estimating VoI is critical for selectively employing open-loop macro-actions because *open-loop actions have bounded regret exactly when VoI is low*.

Before presenting the formal definition of VoI, we give an example to provide intuition about when low VoI may arise in planning problems. A belief may have low VoI when: (i) state dynamics are locally predictable due to the transition function, or (ii) the reward function is insensitive to uncertainty in the current belief, or (iii) sensors are locally uninformative or only infrequent sensing is to necessary to reduce state uncertainty. These conditions are visualized schematically in Fig. 2. These (and other) conditions arise in many real-world planning problems. Consider the problem of a marine robot tracking a plankton bloom using an ocean flow model. State uncertainty grows slowly when the bloom is localized in regions of near-laminar flow (i). Moderate uncertainty in the bloom location may be tolerable when far from human-occupied beaches (ii). Finally, if the state of the bloom can be observed accurately in certain regions of the ocean, only infrequent observation may be necessary; by contrast, in regions where sensor observations are highly noisy, observations may not meaningfully reduce state uncertainty (iii).

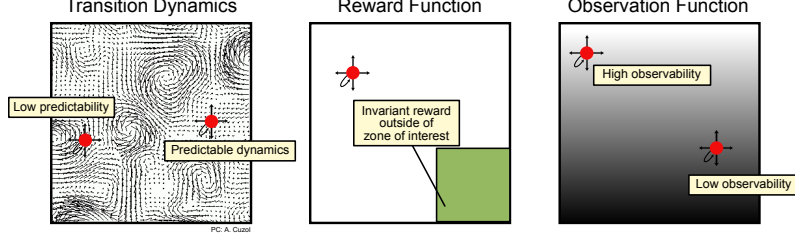

*Figure 2:* ***Conditions for Low VoI****: Value of information may be low in a POMDP when the transition dynamics are locally predictable over short horizons (left), the reward function is invariant to aspects of state uncertainty (middle), or when observations are uninformative due to low state observability (right).*

Rather than specialize an algorithm to recognize conditions (i)-(iii), invariably missing other conditions, we find regions of belief-space where open-loop macro-actions are near-optimal by estimating VoI direclty using the POMDP value function.

### 4.1 Value of Information for Identifying One-Step Open-loop Actions

We adopt a point-based value function representation, *i.e.,* we approximate the value function using a set of $N$ exemplar beliefs $\mathcal{B} = \{b_i\}_{i=0}^{N-1}$. We compute successive value-function approximations to horizon-$H$ using point-based value iteration [15], where backups of beliefs in the set $\mathcal{B}$ leverage a parametric form of the value function over belief space, *e.g.,* a set of $\alpha$-vectors [9] or a deep neural network. We modify the standard value iteration backup operation to compute the VoI, adding open-loop backups whenever the VoI is low. An algorithm summary is presented in the supplement.

We begin by constructing the value function $\hat{V}_h^*$, which approximates the value of a policy that selectively acts in open-loop when VoI is low. $\hat{V}_0^*$ is initialized to the optimal value function. To perform backups of $\hat{V}_h^*$, we compute the open-loop value, $V_h^{OLP}$, which considers acting in open-loop in the current planning iteration:

$$V_h^{OLP}(b) = \max_{a \in \mathcal{A}} \mathbb{E}_{s \sim b}[R(s,a)] + \gamma \hat{V}_{h-1}^*(b^{a,*}) \qquad h = 0, \ldots, H-1, \tag{3}$$

where $b^{a,*}$ represents the open-loop belief transition, marginalizing over the received observation.

During the value function backup, we compute both the standard, closed-loop value (Eq. 2), denoted $V_h^{CLP}$ and the open-loop value $V_h^{OLP}$, where for both backups $\hat{V}_{h-1}^*$ is used to evaluate the recursion.

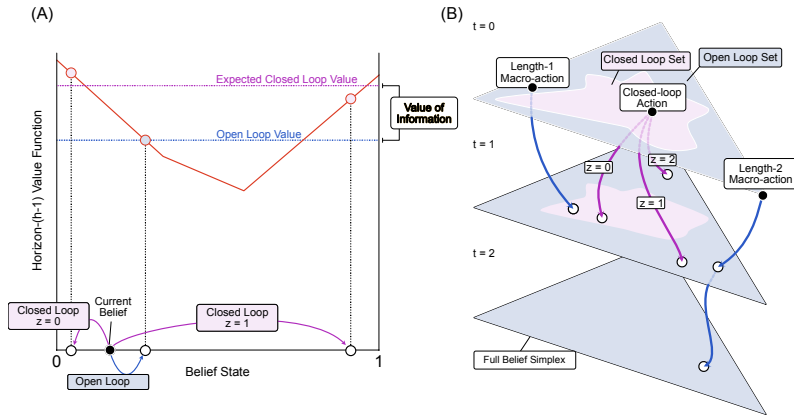

*Figure 3:* ***Macro-action generation****: (A) To compute the value of information at the current belief, we compute immediate reward plus the horizon-$(h-1)$ value under both an open-loop (blue) and closed-loop (purple) belief transition. This difference represents the value of information for long-term task performance. (B) Variable-length macro-actions are constructed by macro-action chaining — for each belief in the open-loop set $\mathcal{B}_h^{OLP}$, we compute the open-loop transition (blue) and terminate macro-action chaining when the belief transitions into the closed-loop set (purple). For beliefs in the closed loop set, no macro-action is generated.*

The difference between the open- and closed-loop value represents the VoI at horizon $h$ for each belief in $\mathcal{B}$ (Figure 3).

$$\text{VoI}_h(b) = V_h^{CLP}(b) - V_h^{OLP}(b) \tag{4}$$

When VoI is below a regret threshold $\tau$ we perform an open-loop backup at $b$. We then add $b$ to the open-loop set $\mathcal{B}_h^{OLP}$ and store the optimal open-loop action in action set $\mathcal{A}_h^{OLP}$. The resulting backup operator is denoted $\hat{\mathcal{H}}$:

$$\hat{V}_h^*(b) = \hat{\mathcal{H}}\hat{V}_{h-1}^*(b) = \begin{cases} V_h^{OLP}(b) & \text{if } V_h^{OLP}(b) \geq V_h^{CLP}(b) - \tau \\ V_h^{CLP}(b) & \text{otherwise} \end{cases} \tag{5}$$

### 4.2 Chaining Open-loop Actions into Macro-Actions

Value iteration is performed to horizon $H$ using Eq. 5, producing the value function $\hat{V}_h^*$, sets of open-loop $\mathcal{B}_h^{OLP}$ beliefs, and optimal open-loop actions $\mathcal{A}_h^{OLP}$. We use these sets to generate near-optimal macro-actions for each belief in $\mathcal{B}$ by performing macro-action chaining (Fig. 3). An algorithm summary for macro-action chaining is presented in the supplement.

Starting from horizon $h$, we iterate over beliefs in $\mathcal{B}$. If $b \notin \mathcal{B}_h^{OLP}$, we immediately terminate macro-action chaining. If $b \in \mathcal{B}_h^{OLP}$, we find the associated open-loop action $a \in \mathcal{A}_h^{OLP}$. Critically, the open-loop belief transition is deterministic conditioned on the selected action. Let $b^{a,*}$ be the belief resulting from open-loop action $a$, which may or may not be included in $\mathcal{B}$. If $b^{a,*} \notin \mathcal{B}$, we evaluate $V_{h-1}^{OLP}(b^{a,*})$ and $V_{h-1}^{CLP}(b^{a,*})$ and use Eq. 5 to decide if $b^{a,*} \in \mathcal{B}_{h-1}^{OLP}$. If so, we find the associated optimal action $a'$ and extend the macro-action chain for belief $b$ to include $a'$; if not, we terminate the chain. We proceed in this manner for the remainder of the planning horizon, or until the deterministically transitioning belief is not in the open-loop set. The chaining process is repeated for each belief in $\mathcal{B}$ and each horizon $h = 0, \ldots, H - 1$.

Macro-action chaining produces a set of belief-dependent macro-actions for beliefs in $\mathcal{B}$. However, during online policy execution, we are likely to encounter beliefs not contained in $\mathcal{B}$. For each belief $b \notin \mathcal{B}$, we execute the macro-action associated with $b$'s nearest neighbor in $\mathcal{B}$ under the $L^1$ norm. Let $V_H^{MA}$ to denote the expected reward of this approximate macro-action policy over horizon $H$.

**Method Summary**   VoI macro-action generation first proceeds backward, performing point-based value iteration to estimate VoI and using VoI to decompose the belief space into beliefs for which an open-loop action is near-optimal (open-loop set) and those for which sensing is needed (closed-loop set) (Section 4.1). Then, macro-action chaining proceeds forward, propagating each belief in the open-loop set forward under the action computed during value iteration and building an open-loop macro-action chain until the propagated belief lies in the closed-loop set (Section 4.2). The resulting VoI macro-actions are belief-dependent and variable-length. This method is visualized in Fig. 3.

## 5 Analysis

In the following, we show that the the macro-action value function $V_H^{MA}$ is within a constant factor of the optimal value function $V_H^*$.

Let $(\mathcal{S}, \mathcal{F}, \mu)$ be a $\sigma$-finite measurable space with $\sigma$-algebra $\mathcal{F}$ and measure $\mu$. Let the belief space $\Pi(\mathcal{S})$ be a subset of $L^1(\mathcal{S}, \mathcal{F}, \mu)$ with $L^1$ norm $\|\cdot\|_1$ and let $\|\cdot\|$ denote the absolute value on $\mathbb{R}$. Let the point-based belief set $\mathcal{B}$ form a $\delta_{\mathcal{B}}$-covering of a compact set $\mathcal{G} \subseteq \Pi(\mathcal{S})$ that contains all beliefs reachable under the VoI macro-action policy $\mathcal{R}^{MA}(b_0) \subseteq \mathcal{G}$. We assume the following:

**Assumption 5.1.** *Let $V_h^\pi$, the horizon $h$ value function under a policy $\pi$ be Lipschitz continuous with Lipschitz constant $L$ over the reachable belief space of initial belief $b_0$ under policy $\pi$: $\|V_h^\pi(b_1) - V_h^\pi(b_2)\| \leq L \|b_1 - b_2\|_1, \forall b_1, b_2 \in \mathcal{R}^\pi(b_0)$.*

This assumption holds for many classes of POMDP problems, including finite-horizon discrete POMDPs [9], finite-horizon continuous POMDPS [13], and information reward POMDPs [5].

Consider the POMDP model $M = (\mathcal{S}, \mathcal{A}, T, R, \mathcal{Z}, O, b_0, H, \gamma)$ with reward bounded in $[-R_{max}, R_{max}]$. We define regret $r_H$ at horizon $H$ as the worst-case difference in long-term expected reward between the optimal policy $V_H^*$ and the approximate VoI macro-action policy $V_H^{MA}$:

**Theorem 5.2.** *The worst-case regret of a policy using VoI macro-actions with threshold $\tau$ is bounded for beliefs in $\mathcal{G}$ by:*

$$r_H = \left\| V_H^* - V_H^{MA} \right\|_\infty \leq \frac{1 - \gamma^H}{1 - \gamma} \left( \delta_{\mathcal{B}} \left( 3L + \frac{R_{max}}{1 - k\gamma} + L\gamma k \right) + \tau \right). \tag{6}$$

We prove Theorem 5.2 in the remainder of this section. The proof relies on two key results: Lemma 5.3, which bounds value approximation error during VoI-based backups and Lemma 5.5, which bounds the error of applying macro-actions computed offline to beliefs at runtime.

## 5.1 Value Backup Error

Let $\mathcal{H}$, $V_h^*$ denote the exact value backup operator and the resulting optimal value function respectively (Eq. 2) and $\hat{\mathcal{H}}$, $\hat{V}_h^*(b)$ denote the point-based, macro-action backup and value function (Eq. 5), where the optimal open-loop action for each belief is used without macro-action chaining. We show that the error between $V_h^*$ and $\hat{V}_h^*$ is bounded and can be decomposed into error caused by the point-based approximation and error caused by the inclusion of potentially sub-optimal open-loop actions.

**Lemma 5.3.** *The horizon-$H$ value function error caused by including open-loop actions in backups whenever VoI $< \tau$ is bounded for beliefs in $\mathcal{G}$ by $\epsilon_H = \left\| \hat{V}_H^* - V_H^* \right\|_\infty \leq \frac{1 - \gamma^H}{1 - \gamma}(2L\delta_{\mathcal{B}} + \tau)$.*

This bound illuminates the role of VoI parameter $\tau$ in policy performance — for larger $\tau$, open-loop actions are used frequently and $\delta_{\mathcal{B}}$ will generally decrease, improving policy regret; however, large $\tau$ also contributes to the policy regret by allowing open-loop actions be taken even when VoI is high.

*Proof.* Consider any compact subset $\beta$ of $\mathcal{G}$; importantly, $\beta$ can be different from the belief set $\mathcal{B}$ used in planning. We define $\epsilon_h$ to be the maximum error in the value function on the set $\beta$ during the value iteration recursion at horizon $h$. Let $b_\epsilon \in \beta$ be the belief for which the value function error is maximized and $\delta$ be the minimum distance between a belief in $\mathcal{B}$ and $b_\epsilon$: $\delta = \min_{b \in \mathcal{B}} \|b - b_\epsilon\|_1$. Because $\mathcal{B}$ forms a $\delta_{\mathcal{B}}$ covering of $\beta$, we have that $\delta \leq \delta_{\mathcal{B}}$. We bound $\epsilon_h$ (proof in the supplement, Section B) by the following term: $\epsilon_h = \left\| V_h^*(\beta) - \hat{V}_h^*(\beta) \right\|_\infty \leq 2L\delta_{\mathcal{B}} + \left\| \mathcal{H} V_{h-1}^*(b) - \hat{\mathcal{H}} \hat{V}_{h-1}^*(b) \right\|$.

The term $2L\delta_{\mathcal{B}}$ represents the value-function error induced by the point-based approximation [15]. We will further examine the term $\mathcal{H} V_{h-1}^*(b) - \hat{\mathcal{H}} \hat{V}_{h-1}^*(b)$. Without loss of generality, let $a_1$ be the optimal, closed-loop action at belief $b$ and $a_2$ be the near-optimal, open-loop action selected for backing up $\hat{V}_h^*$. Let $\mathcal{H}^{a_1}$ denote the closed-loop value function backup using action $a_1$ and $\hat{\mathcal{H}}^{a_2,OLP}$ denote the open-loop backup using action $a_2$.

$$\left\| \mathcal{H} V_{h-1}^*(b) - \hat{\mathcal{H}} \hat{V}_{h-1}^*(b) \right\| = \left\| \mathcal{H}^{a_1} V_{h-1}^*(b) - \hat{\mathcal{H}}^{a_2,OLP} \hat{V}_{h-1}^*(b) \right\|, \tag{7}$$

$$\leq \left\| \mathcal{H}^{a_2,OLP} V_{h-1}^*(b) + \tau - \hat{\mathcal{H}}^{a_2,OLP} \hat{V}_{h-1}^*(b) \right\|, \tag{8}$$

$$\leq \left\| \gamma V_{h-1}^*(b^{a_2,*}) + \tau - \gamma \hat{V}_{h-1}^*(b^{a_2,*}) \right\| \leq \gamma \epsilon_{h-1} + \tau, \tag{9}$$

where if $b^{a_2,*} \notin \mathcal{G}$, we replace $V_{h-1}^*(b^{a_2,*})$, $\hat{V}_{h-1}^*(b^{a_2,*})$ with a valid lower-bound. Expanding the recursion $\epsilon_h \leq \gamma \epsilon_{h-1} + 2L\delta_{\mathcal{B}} + \tau$, we conclude that $\epsilon_H \leq \frac{1 - \gamma^H}{1 - \gamma}(2L\delta_{\mathcal{B}} + \tau)$. $\qquad \square$

## 5.2 Generalizing Macro-Actions

During policy execution, we generalize macro-actions computed for beliefs in $\mathcal{B}$ to new beliefs. The error induced by this approximation can be bounded by demonstrating that the open-loop dynamics are a non-expansive mapping in belief-space, ensuring that during an open-loop macro-action, the distance between the forward-propagated beliefs can be no larger than their initial separation $\delta$.

**Lemma 5.4.** *(Lasota and Mackey [12]) The open-loop dynamics are a non-expansive mapping in belief space. Consider two beliefs $b_1, b_2 \in \Pi(\mathcal{S})$ such that $\|b_1 - b_2\|_1 = \delta$. Then, for any action $a$ taken in open-loop, it follows that $\left\| b_1^{a,*} - b_2^{a,*} \right\|_1 \leq k\delta$ for $0 \leq k \leq 1$.*

Let $V_h^{MA}$ denote the value of following an approximate macro-action policy from belief $b$ at horizon $h$, where $b \notin \mathcal{B}$ and the macro-action computed for its nearest neighbor $b_*$ is instead executed.

**Lemma 5.5.** *The additional value function error of approximating the VoI macro-action at belief $b$ using its nearest neighbor $b_*$ under $k$-contractive open-loop dynamics is bounded by:*

$$\eta_H = \left\| \hat{V}_H^* - V_H^{MA} \right\|_\infty \leq \frac{1 - \gamma^H}{1 - \gamma} \left( L\delta_{\mathcal{B}} + \frac{R_{max}\delta_{\mathcal{B}}}{1 - \gamma k} + L\gamma k\delta_{\mathcal{B}} \right). \tag{10}$$

*Proof.* Consider any compact subset $\beta$ of $\mathcal{G}$. Then $\eta_h = \left\| \hat{V}_h^*(\beta) - V_h^{MA}(\beta) \right\|_\infty$.

Without loss of generality, let $b$ be the belief for which $\eta_h$ is maximized, let $b_*$ be its nearest neighbor in $\mathcal{B}$, and let $A_l = \{a_1, \ldots, a_l\}$ be the length-$l$ macro-action that is optimal at $b_*$. We show (details in the supplement, Section B) that $\eta_h$ can be decomposed into error incurred during the macro-action and error over the remainder of the planning horizon. We use the notation $b^{A_{1:i}}$ to denote an updated belief after taking the first $i$ actions of macro-action $A_l$ in open-loop.

$$\eta_h \leq L\delta_{\mathcal{B}} + \left\| \sum_{i=0}^{l-1} \gamma^i \Big( \mathbb{E}_{s \sim b_*^{A_{1:i}}}[R(s, a_i)] - \mathbb{E}_{s \sim b^{A_{1:i}}}[R(s, a_i)] \Big) \right\| + \gamma^l (Lk^l\delta_{\mathcal{B}} + \eta_{h-l}). \tag{11}$$

The form of Eq. 11 reflects the expected reward when following the macro-action $A_l$ from both belief $b$ and $b_*$ and then reverting to the macro-action policy from the resulting belief. We bound Eq. 11 by application of the non-expansive property:

$$\left\| \sum_{i=0}^{l-1} \gamma^i \Big( \mathbb{E}_{s \sim b_*^{A_{1:i}}}[R(s, a_i)] - \mathbb{E}_{s \sim b^{A_{1:i}}}[R(s, a_i)] \Big) \right\|, \tag{12}$$

$$\leq \sum_{i=0}^{l-1} \gamma^i R_{max} \left\| b_*^{A_{1:i}} - b^{A_{1:i}} \right\|_1 \leq \sum_{i=0}^{l-1} \gamma^i R_{max} k^i \left\| b_* - b \right\|_1 \leq \frac{1 - \gamma^l k^l}{1 - \gamma k} R_{max}\delta. \tag{13}$$

Plugging this expression into Eq. 11, we have the recursion: $\eta_h \leq L\delta_{\mathcal{B}} + \frac{1 - \gamma^l k^l}{1 - \gamma k} R_{max}\delta_{\mathcal{B}} + \gamma^l Lk^l\delta_{\mathcal{B}} + \gamma^l \eta_{h-l}$. This expression depends on $l$, the length of the optimal macro-action at horizon $h$, in a a complex way. Because $l$ is variable and unknown *a priori*, we replace $l$ with its worst-case value in each expression: $\eta_h \leq L\delta_{\mathcal{B}} + \frac{R_{max}\delta_{\mathcal{B}}}{1 - \gamma k} + \gamma Lk\delta_{\mathcal{B}} + \gamma\eta_{h-1}$ and expand the recursion. $\square$

**Analysis Summary** To bound the regret of the VoI macro-action policy compared to the optimal policy, we first bound the error caused by using sub-optimal open-loop actions when VoI was below a threshold $\tau$. We then bound the regret of generalizing macro-actions generated for beliefs in $\mathcal{B}$ to new beliefs encountered during policy execution. Finally, we combine these approximation errors:

*Proof.* *(Theorem 5.2)* We bound the regret of the VoI macro-action policy on the set $\mathcal{G}$ as follows:

$$r_H = \left\| V_H^* - V_H^{MA} \right\|_\infty \leq \left\| V_H^* - \hat{V}_H^* \right\|_\infty + \left\| \hat{V}_H^* - V_H^{MA} \right\|_\infty = \epsilon_H + \eta_H. \tag{14}$$

The result follows by applying Lemma 5.3 and Lemma 5.5 to bound $\epsilon_H$ and $\eta_H$. $\square$

## 6 Experiments

We present experimental results designed to highlight various aspects of VoI macro-actions and provide insight into the nature of the regret bound and its implications macro-action design. We assume discrete states, actions and observations and represent the value function by a piecewise-linear and convex (PWLC) collection of $\alpha$-vectors [9]. An adaptation of the algorithm presented in Section 4 to a PWLC value function is contained in the supplement (Section C).

We demonstrate macro-action generation in a dynamic tracking problem (Fig. 4), in which a fully observable, actuated agent tracks a partially observable target moving in a known $10 \times 10$ discretized

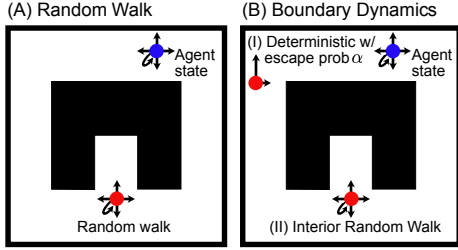

(A) Random Walk — Agent state — Random walk

(B) Boundary Dynamics — (I) Deterministic w/ escape prob $\alpha$ — Agent state — (II) Interior Random Walk

*Figure 4: An agent (blue) tracks a partially-observable target (red) in an environment with obstacles and boundaries (black) under (A) a random walk or (B) a boundary dynamic with escape probability $\alpha$.*

| Planner | Total Reward | Empirical $\delta_{\mathcal{B}}$ |
|---|---|---|
| Boundary Dynamics, $\alpha = 0.20$ | | |
| Base CL | 2800.2 (465.6)* | 0.45 (0.04)* |
| **VoI MA** | **3039.5 (274.4)** | **0.29 (0.03)** |
| Fixed MA | 2584.3 (460.5)* | 0.17 (0.02)* |
| Random Walk Dynamics | | |
| Base CL | 3035.8 (257.5) | 0.42 (0.04)* |
| **VoI MA** | **3050.8 (212.9)** | **0.31 (0.04)** |
| Fixed MA | 2946.6 (311.4)* | 0.15 (0.02)* |

*Table 1: Realized reward (higher is better) and empirical estimates of $\delta_{\mathcal{B}}$ (lower is better) during $M = 500$ experiments using VoI MA, Base CL, and Fixed MA policies (mean, std). Significant differences from VoI MA (two-sided Welch's t-test with Bonferroni correction, $p < 0.05/2$) are indicated with an asterisk.*

map ($|\mathcal{S}| = 10,000$). A full description of the experimental domain and parameterization is in the supplement (Section D). We use this example to expose several key low VoI regimes caused by structure in the system dynamics, reward function, and transition dynamics.

Our first experiment is designed to highlight how locally predictable state dynamics can lead to low VoI values (condition (i), Section 4). We test both a *random-walk* target dynamic and a *boundary* target dynamic in which the target performs a random walk in the interior but moves deterministically clockwise on the boundaries, with probability $\alpha = 0.20$ of returning to the interior (Fig. 4). These dynamics exemplify different VoI regimes. In the boundary dynamic, VoI is low on the boundary, allowing for long macro-actions. In the random walk dynamic, VoI is more uniform over belief space; only short macro-actions are possible before sensing is necessary.

Our second experiment introduces conditions (ii) and (iii) (Section 4) by rewarding the agent for tracking the target only in a single zone of interest in the upper left corner of the world and imposing non-uniform observation noise inspired by the Dark-Light POMDP problem [16, 20], such that the agent can sense the target's location most accurately on the bottom half of the world (Fig. 6). The agent follows a boundary transition dynamic (condition (i)). The agent must perform information gathering – moving to the bottom of the world to localize the target — before returning to the upper corner to track the target in the zone of interest. VoI is high when the target is nearing the zone of interest and can be localized with sufficient accuracy via sensing.

The VoI macro-action policy (VoI MA) is compared against an approximation to the closed-loop optimal policy (*base closed-loop*, Base CL) and a *fixed length macro-action* (Fixed MA) policy, which is constrained to act closed-loop only every $T = 15$ planning iterations. For all three policies, value function approximation is performed using a custom implementation of PBVI [15]; the details of value function approximation can be found in the supplement (Section D).

## 6.1 Experimental Results

Results for Experiment 1 are shown in Table 1. Under the boundary dynamic, the VoI macro-action policy has higher cumulative reward than the Base CL and Fixed MA policies. This may seem counterintuitive — the performance of the optimal policy is an upper bound on the VoI macro-action policy. However, this is an example of the trade-off between policy complexity and approximability indicated by Theorem 5.2. The observed value of $\delta_{\mathcal{B}}$ is significantly lower for the VoI macro-action policy (Table 1), indicative of the macro-action policy's smaller reachable belief space. We note that the fixed macro-action policy has the smallest value of $\delta_{\mathcal{B}}$. However, the fixed-length macro-actions, like many other hand-coded macro-actions, can be arbitrarily sub-optimal. For the random walk dynamic, there is less opportunity to exploit open-loop actions and we see that, as we would hope, VoI MA policy performance reverts to that of the closed-loop policy.

We additionally explore the effect of the VoI threshold $\tau$ on planner performance, macro-action utilization, and the value of $\delta_{\mathcal{B}}$. Results are presented in Fig. 5. As $\tau$ increases, the VoI macro-action policy acts in open-loop for a larger fraction of the planning horizon and the value of $\delta_{\mathcal{B}}$ decreases.

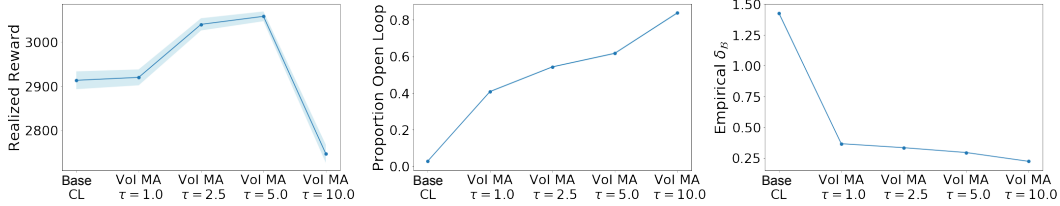

*Figure 5: (Left) Realized reward under Base CL and VoI macro-action policies with increasing values of the VoI threshold $\tau$. (Center) The proportion of the planning horizon for which open-loop macro-actions are employed. (Right) The empirical value of $\delta_{\mathcal{B}}$. Plots show mean and standard error in $M = 500$ trials.*

The realized reward reflects the balance between these two terms, initially increasing as $\delta_{\mathcal{B}}$ decreases, before finally decreasing as the policy incorporates more sub-optimal open-loop actions.

For Experiment 2, we visualize the length of the discovered belief-dependent macro-actions. To visualize macro-actions as a function of the high-dimensional belief space, for each possible state in the world, we compute the length of the macro-action corresponding to a belief where the the target is localized to that state (the target probability mass function is a delta function), averaged over all possible corresponding states of the agent. Results are shown in Fig. 6. The effect of structure in the POMDP model on the discovered macro-actions is evident — when the target is localized in the zone of interest (upper left corner) or has just passed the zone and is moving clockwise due to the boundary dynamic, the agent can achieve high reward with open loop macroactions; when the target is in the lower half of the world and moving towards the zone of interest, sensing is crucial. In this experiment, the VoI MA policy achieves higher average reward (18.69) than the Base CL (15.08) and Fixed MA (13.03) policies over $M = 100$ simulated trails.

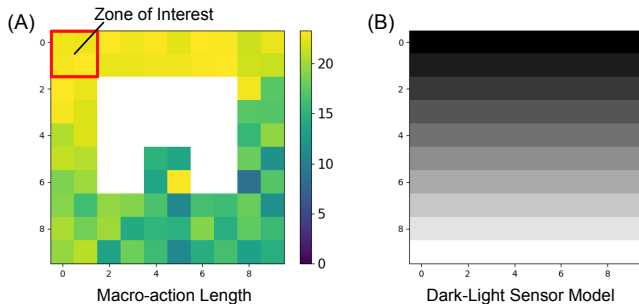

*Figure 6: **Visualizing Belief-Dependent Macro-Actions.** (A) For each state in the world, the average length of the discovered VoI macro-action is shown for each possible state of the target. If the target is localized near the upper-left corner of the world, the agent can confidently track the target in the zone of interest, and long-horizon macro-actions are possible. Sensing is more important in the bottom half of the world; the agent must move to the higher observability regions near the bottom of the world in order to reduce target uncertainty before it enters the zone of interest. VoI macro-actions are also sensitive to the state dynamics — for example, long-horizon macro-actions are possible when the target is trapped in the bugtrap obstacle. (B) The dark-light sensor model; sensor noise increases linearly from the bottom (light) to the top (dark) of the world.*

# 7 Conclusion

This work presents value-of-information macro-actions for planning under uncertainty. By generating macro-actions using VoI, we bound the regret of macro-action policies with respect to the optimal, closed-loop policy. Leveraging open-loop macro-actions within POMDP policies can reduce the size of a policy's reachable belief space and thus the complexity of planning. This has direct implications for the performance of point-based POMDP policies, as we show theoretically in Theorem 5.2 and experimentally in a set of dynamic tracking experiments. VoI macro-actions balance the planning complexity induced by sensing with the value of the information provided by observations in partially-observable environments to enable efficient task execution in POMDPs.

## Broader Impact

Decision-making problems are ubiquitous, arising in applications such as tracking an oil spill using a marine robot, selecting an effective drug schedule in personalized medicine, or allocating irrigation resources based on seasonal weather forecasts. In each of these important application areas, system dynamics are represented by complex and potentially learned models and the decision-making agent can only observe the state through limited sensors. Many current planning and reinforcement learning algorithms focus on fully-observable domains and generate learned policies without performance guarantees. However, uncertainty and formal guarantees must play a role in robust decision-making for high-stakes domains. VoI macro-action generation contributes to fundamental research in robust and efficient model-based planning under uncertainty. As with all formal results, however, the bounds we derive only hold under the assumptions that we describe in the text. When performing decision-making in high-stakes applications, understanding these conditions, the extent to which they hold, and how algorithm performance degrades as assumptions are violated is critical.

## Acknowledgments and Disclosure of Funding

The authors would like to thank the annonymous reviwers for their thoughtful feedback, and the members of Sensing, Learning and Inference (SLI) group and the Robust Robotics Group (RRG) for many insightful discussions. This research was partially supported by ONR (Award No. N00014-17-1-2072), DOE/NNSA (Award No. DE-NA0003921), DARPA (Award No. HR001120C0033), and an NSF Graduate Research Fellowship (GF).

## Footnotes

[1] We use the term *macro-action* synonymously with open loop action sequence.

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
