[Supplementary Material]

# Belief-Dependent Macro-Action Discovery in POMDPs using the Value of Information

**Genevieve Flaspohler**[1,2]**, Nicholas Roy**[1]**, and John W. Fisher III**[1]

Massachusetts Intitute of Technology[1] and the Woods Hole Oceanographic Institution[2]

{geflaspo, nickroy, fisher}@csail.mit.edu

This supplement includes additional details, figures, and analysis not presented in the main text due to space limitations. Section A presents algorithm descriptions for macro-action generation, including modified value iteration and macro-action chaining. Section B includes detailed derivations of Lemma 5.3, 5.4 and 5.5 presented in the main text. Section C presents the details of macro-action generation in the case of discrete POMDPs with $\alpha$-vector value function representations and discusses the algorithmic complexity of macro-action generation in the discrete case. Finally, Section D provides additional visualizations and discussion of experimental results.

## A   Macro-action Generation

VoI macro-action generation first proceeds backward, performing point-based value iteration to decompose the belief set $\mathcal{B}$ into beliefs for which an open-loop action is near-optimal and for which a sensor observation is needed. Algorithm 1 details the modified value iteration procedure introduced in the main text.

The belief set $\mathcal{B}$ is initialized at the start of the algorithm by the `construct_belief_set` (e.g., using random beliefs or beliefs reachable under a QMDP policy [4]). As is standard in point-based POMDP literature [5], we perform iterations of alternating value-iteration and belief set updates, in which the value function estimate is used to update the set of reachable beliefs $\mathcal{B}$, via the `expand_belief_set` method, which in turn is used to improve the value function estimate. The algorithm input is a POMDP model, a VoI threshold $\tau$, and a number of belief-set update iterations $iters$.

After value iteration, macro-action chaining proceeds forward, propagating each belief in the open-loop set forward under the optimal action computed during value iteration, and returning when the propagated belief lies in the closed-loop set. Algorithm 2 details the procedure for macro-action chaining. Macro-action chaining takes as input the belief set $\mathcal{B}$, and the VoI macro-action value function $\{\hat{V}_h^*\}_{h=0}^{H-1}$, open-loop sets $\{\mathcal{B}_h^{OLP}\}_{h=0}^{H-1}$, and optimal open-loop actions $\{\mathcal{A}_h^{OLP}\}_{h=0}^{H-1}$. The algorithm checks if a belief $b$ is in the open-loop set at horizon $h$; for beliefs $b \notin \mathcal{B}$, `check_open_loop` computes VoI as described in Algorithm 1 to determine if an open-loop action is near-optimal at $b$.

---

**Algorithm 1** Value Iteration for Macro-Action Planning

---

**Input:** POMDP $= (\mathcal{S}, \mathcal{A}, T, R, \mathcal{Z}, O, b_0, H, \gamma), iters, \tau$
$\mathcal{B} = \texttt{construct\_belief\_set}(b_0)$
**for** $i = 0, \ldots, iters - 1$ **do**
    $\hat{V}_0^*(b) = \max_{a \in \mathcal{A}} \mathbb{E}_{s \sim b}[R(s, a)], \forall b \in \mathcal{B}$
    $\mathcal{B}_0^{OLP} = \mathcal{B}$ // Open loop actions are optimal at horizon-0
    **for** $h = 1, \ldots, H - 1$ **do**
        **for** $b \in \mathcal{B}$ **do**
            $V_h^{OLP}(b) = \max_{a \in \mathcal{A}} \mathbb{E}_{s \sim b}[R(s, a)] + \gamma \hat{V}_{h-1}^*(b^{a,*})$
            $V_h^{CLP}(b) = \max_{a \in \mathcal{A}} \mathbb{E}_{s \sim b}[R(s, a)] + \gamma \int_{\mathcal{Z}} P(z \mid b, a) \hat{V}_{h-1}^*(b^{a,z}) \mathrm{d}z$
            $\mathrm{VoI}_h(b) = V_h^{CLP}(b) - V_h^{OLP}(b)$
            **if** $\mathrm{VoI}_h(b) \leq \tau$ **then**
                Add $b$ to $\mathcal{B}_h^{OLP}$ and optimal action to $\mathcal{A}_h^{OLP}$.
                $\hat{V}_h^*(b) = V_h^{OLP}(b)$
            **else**
                $\hat{V}_h^*(b) = V_h^{CLP}(b)$
            **end if**
        **end for**
    **end for**
    $\mathcal{B} = \texttt{expand\_belief\_set}(b_0, \mathcal{B}, \{\hat{V}_h^*\}_{h=0}^{H-1}, \{\mathcal{B}_h^{OLP}\}_{h=0}^{H-1})$
**end for**
**return** $\mathcal{B}, \{\hat{V}_h^*\}_{h=0}^{H-1}, \{\mathcal{B}_h^{OLP}\}_{h=0}^{H-1}, \{\mathcal{A}_h^{OLP}\}_{h=0}^{H-1}$

---

---

**Algorithm 2** Macro-Action Chaining

---

**Input:** $\mathcal{B}, \{\hat{V}_h^*\}_{h=0}^{H-1}, \{\mathcal{B}_h^{OLP}\}_{h=0}^{H-1}, \{\mathcal{A}_h^{OLP}\}_{h=0}^{H-1}$
$\texttt{macroaction}[b, 0] = \mathcal{A}_0^{OLP}(b), \forall b \in \mathcal{B}$ // Initialize horizon zero macro-actions
**for** $h = 1, \ldots, H - 1$ **do**
    **for** $b \in \mathcal{B}$ **do**
        $b' = b$
        **for** $t = h, \ldots, 0$ **do**
            //Check if $b'$ is in the open-loop set and, if so, get open-loop action $a^*$.
            $\texttt{olp}, a^* = \texttt{check\_open\_loop}(b', t, \mathcal{B}, \{\hat{V}_h^*\}_{h=0}^{H-1}, \{\mathcal{B}_h^{OLP}\}_{h=0}^{H-1})$
            **if** not $\texttt{olp}$ **then**
                break // Sensor observation necessary, terminate macro-action chaining
            **end if**
            $\texttt{macroaction}[b, h].\texttt{append}(a^*)$
            $b' = b'^{a^*,*}$ // Transition belief according to open-loop dynamics
        **end for**
    **end for**
**end for**

---

# B  Analysis

The following section contains the detailed derivation of Lemmas 5.3-5.5 presented in the main text.

## B.1  Value Backup Error

**Lemma 5.3**  *The horizon-$H$ value function error caused by including open-loop actions in backups whenever VoI $< \tau$ is bounded for beliefs in $\mathcal{G}$ by $\epsilon_H = \left\| \hat{V}_H^* - V_H^* \right\|_\infty \leq \frac{1-\gamma^H}{1-\gamma}(2L\delta_\mathcal{B} + \tau)$.*

*Proof.* Consider any compact subset $\beta$ of $\mathcal{G}$; importantly, $\beta$ can be different from the belief set $\mathcal{B}$ used in planning. We define $\epsilon_h$ to be the maximum error in the value function on the set $\beta$ during the value iteration recursion at horizon $h$:

$$\epsilon_h = \left\| V_h^*(\beta) - \hat{V}_h^*(\beta) \right\|_\infty, \tag{S1}$$

where we use the notation $f(\beta)$ to denote the restriction of the function $f$ to the domain $\beta$. Then:

$$\epsilon_h = \left\| \mathcal{H}V_{h-1}^*(\beta) - \hat{\mathcal{H}}\hat{V}_{h-1}^*(\beta) \right\|_\infty, \tag{S2}$$

Let $b_\epsilon \in \beta$ be the belief for which the value function error is maximized:

$$b_\epsilon = \arg\max_{b \in \beta} \left\| \mathcal{H}V_{h-1}^*(b) - \hat{\mathcal{H}}\hat{V}_{h-1}^*(b) \right\|. \tag{S3}$$

Let $\delta$ be the minimum distance between a belief in $\mathcal{B}$ and $b_\epsilon$: $\delta = \min_{b \in \mathcal{B}} \|b - b_\epsilon\|_1$ and let belief $b \in \mathcal{B}$ be a minimizer. Because $\mathcal{B}$ forms a $\delta_\mathcal{B}$ covering of $\beta$, we have that $\delta \leq \delta_\mathcal{B}$.

We bound $\epsilon_h$ as follows:

$$\epsilon_h = \left\| V_h^*(\beta) - \hat{V}_h^*(\beta) \right\|_\infty \tag{S4}$$

$$= \left\| \mathcal{H}V_{h-1}^*(b_\epsilon) - \hat{\mathcal{H}}\hat{V}_{h-1}^*(b_\epsilon) \right\| \tag{S5}$$

$$= \left\| \mathcal{H}V_{h-1}^*(b_\epsilon) - \mathcal{H}V_{h-1}^*(b) + \mathcal{H}V_{h-1}^*(b) - \hat{\mathcal{H}}\hat{V}_{h-1}^*(b_\epsilon) + \hat{\mathcal{H}}\hat{V}_{h-1}^*(b) - \hat{\mathcal{H}}\hat{V}_{h-1}^*(b) \right\| \tag{S6}$$

$$\leq \left\| \mathcal{H}V_{h-1}^*(b_\epsilon) - \mathcal{H}V_{h-1}^*(b) \right\| + \tag{S7}$$
$$\left\| \hat{\mathcal{H}}\hat{V}_{h-1}^*(b_\epsilon) - \hat{\mathcal{H}}\hat{V}_{h-1}^*(b) \right\| + \left\| \mathcal{H}V_{h-1}^*(b) - \hat{\mathcal{H}}\hat{V}_{h-1}^*(b) \right\|$$

$$\leq 2L \|b_\epsilon - b\|_1 + \left\| \mathcal{H}V_{h-1}^*(b) - \hat{\mathcal{H}}\hat{V}_{h-1}^*(b) \right\|. \tag{S8}$$

$$\leq 2L\delta_\mathcal{B} + \left\| \mathcal{H}V_{h-1}^*(b) - \hat{\mathcal{H}}\hat{V}_{h-1}^*(b) \right\|. \tag{S9}$$

The term $2L\delta_\mathcal{B}$ represents the value-function error induced by the point-based approximation. We will further examine the term $\mathcal{H}V_{h-1}^*(b) - \hat{\mathcal{H}}\hat{V}_{h-1}^*(b)$, which represents the value function error due to the inclusion of potentially suboptimal macro-actions in the approximate backup operator.

Without loss of generality, let $a_1$ be the optimal action at belief $b$ and $a_2$ be the near-optimal, open-loop action selected for backing up $\hat{V}_h^*$. Let $\mathcal{H}^{a_1}$ denote the standard value function backup using action $a_1$ and $\hat{\mathcal{H}}^{a_2,OLP}$ denote the macro-action backup using action $a_2$ in open loop.

$$\left\|\mathcal{H}V_{h-1}^*(b) - \hat{\mathcal{H}}\hat{V}_{h-1}^*(b)\right\| = \left\|\mathcal{H}^{a_1}V_{h-1}^*(b) - \hat{\mathcal{H}}^{a_2,OLP}\hat{V}_{h-1}^*(b)\right\| \tag{S10}$$

$$\leq \left\|\mathcal{H}^{a_2}V_{h-1}^*(b) + \tau - \hat{\mathcal{H}}^{a_2,OLP}\hat{V}_{h-1}^*(b)\right\| \tag{S11}$$

$$\leq \left\|\mathcal{H}^{a_2,OLP}V_{h-1}^*(b) + \tau - \hat{\mathcal{H}}^{a_2,OLP}\hat{V}_{h-1}^*(b)\right\| \tag{S12}$$

$$\leq \|\mathbb{E}_{s\sim b}[R(s,a_2)] + \gamma V_{h-1}^*(b^{a_2,*}) + \tau - \tag{S13}$$

$$\mathbb{E}_{s\sim b}[R(s,a_2)] - \gamma \hat{V}_{h-1}^*(b^{a_2,*})\|$$

$$\leq \left\|\gamma V_{h-1}^*(b^{a_2,*}) + \tau - \gamma \hat{V}_{h-1}^*(b^{a_2,*})\right\| \tag{S14}$$

$$\leq \gamma\epsilon_{h-1} + \tau, \tag{S15}$$

where if $b^{a_2,*} \notin \mathcal{G}$, we replace $V_{h-1}^*(b^{a_2,*})$, $\hat{V}_{h-1}^*(b^{a_2,*})$ with a valid lower-bound.

Because $V_0^* \equiv \hat{V}_0^*$, we have that $\epsilon_0 = 0$. Expanding the recursion $\epsilon_h \leq \gamma\epsilon_{h-1} + 2L\delta_{\mathcal{B}} + \tau$, we conclude that $\epsilon_H \leq \frac{1-\gamma^H}{1-\gamma}(2L\delta_{\mathcal{B}} + \tau)$. □

### B.2 Interpolating Macro-Actions

**Lemma 5.4** *(Lasota and Mackey [3]) The open-loop dynamics are a non-expansive mapping in belief space. Consider two beliefs $b_1, b_2 \in \Pi(\mathcal{S})$ such that $\|b_1 - b_2\|_1 = \delta$. Then, for any action $a$ taken in open-loop, it follows that $\left\|b_1^{a,*} - b_2^{a,*}\right\|_1 \leq k\delta$ for $0 \leq k \leq 1$.*

*Proof.* Following [3], let $T_a$ be the Markov operator corresponding to the POMDP open-loop transition dynamics under action $a$:

$$\left\|b_1^{a,*} - b_2^{a,*}\right\|_1 = \|T_ab_1 - T_ab_2\|_1 \tag{S16}$$

$$= \|T_a(b_1 - b_2)\|_1 \tag{S17}$$

$$\leq k\|b_1 - b_2\|_1 = k\delta, \tag{S18}$$

where $0 \leq k \leq 1$ is the maximum contraction coefficient over actions $a \in \mathcal{A}$ and Eq. S18 is a property of Markov operators [3]. □

**Lemma 5.5** *The additional value function error of approximating the VoI macro-action at belief $b$ using its nearest neighbor $b_*$ under $k$-contractive open-loop dynamics is bounded by:*

$$\eta_H = \left\|\hat{V}_H^* - V_H^{MA}\right\|_\infty \leq \frac{1-\gamma^H}{1-\gamma}\left(L\delta_{\mathcal{B}} + \frac{R_{max}\delta_{\mathcal{B}}}{1-\gamma k} + L\gamma k\delta_{\mathcal{B}}\right). \tag{S19}$$

*Proof.* Consider any compact subset $\beta$ of $\mathcal{G}$. Define $\eta_h$ to be the maximum difference between the value when utilizing optimal open-loop actions when VoI is low $\hat{V}_h^*$ and the value when performing macro-action chaining and interpolation $V_h^{MA}$:

$$\eta_h = \left\|\hat{V}_h^*(\beta) - V_h^{MA}(\beta)\right\|_\infty \tag{S20}$$

Without loss of generality, let $b$ be the belief for which Eq. S20 is maximized, let $b_*$ be it's nearest neighbor in $\mathcal{B}$, and let $A_l = \{a_1, \ldots, a_l\}$ be the length-$l$ macro-action that is optimal at $b_*$.

$$\eta_h = \left\| \hat{V}_h^*(b) - V_h^{MA}(b) \right\|, \tag{S21}$$

$$\leq \left\| \hat{V}_h^*(b) - \hat{V}_h^*(b_*) \right\| + \left\| \hat{V}_h^*(b_*) - V_h^{MA}(b) \right\|, \tag{S22}$$

$$\leq L\delta_{\mathcal{B}} + \left\| \sum_{i=0}^{l-1} \gamma^i \mathbb{E}_{s \sim b_*^{A_{1:i}}}[R(s,a_i)] + \gamma^l \hat{V}_{h-l}^*(b_*^{A_{1:l}}) \right. \tag{S23}$$

$$\left. - \sum_{i=0}^{l-1} \gamma^i \mathbb{E}_{s \sim b^{A_{1:i}}}[R(s,a_i)] - \gamma^l V_{h-l}^{MA}(b^{A_{1:l}}) \right\|,$$

$$\leq L\delta_{\mathcal{B}} + \left\| \sum_{i=0}^{l-1} \gamma^i \Big( \mathbb{E}_{s \sim b_*^{A_{1:i}}}[R(s,a_i)] - \mathbb{E}_{s \sim b^{A_{1:i}}}[R(s,a_i)] \Big) \right\| \tag{S24}$$

$$+ \gamma^l \left\| \hat{V}_{h-l}^*(b_*^{A_{1:l}}) - V_{h-l}^{MA}(b^{A_{1:l}}) \right\|.$$

$$\leq L\delta_{\mathcal{B}} + \left\| \sum_{i=0}^{l-1} \gamma^i \Big( \mathbb{E}_{s \sim b_*^{A_{1:i}}}[R(s,a_i)] - \mathbb{E}_{s \sim b^{A_{1:i}}}[R(s,a_i)] \Big) \right\| \tag{S25}$$

$$+ \gamma^l \left( \left\| \hat{V}_{h-l}^*(b_*^{A_{1:l}}) - \hat{V}_{h-l}^*(b^{A_{1:l}}) \right\| + \left\| \hat{V}_{h-l}^*(b^{A_{1:l}}) - V_{h-l}^{MA}(b^{A_{1:l}}) \right\| \right)$$

$$\leq L\delta_{\mathcal{B}} + \left\| \sum_{i=0}^{l-1} \gamma^i \Big( \mathbb{E}_{s \sim b_*^{A_{1:i}}}[R(s,a_i)] - \mathbb{E}_{s \sim b^{A_{1:i}}}[R(s,a_i)] \Big) \right\| + \gamma^l (Lk^l \delta_{\mathcal{B}} + \eta_{h-l}) \tag{S26}$$

where Eqs. S22 and S25 follow by the triangle inequality, Eq. S23 using the fact that $A_l$ is the optimal macro-action for $b_*$ and will be applied to $b$ under the macro-action policy, and Eq. S26 by the contractive property of the open loop dynamics.

The form of Eq. S23 reflects the expected reward when following the macro-action $A_l$ from both belief $b$ and $b_*$ and then reverting to the macro-action policy for the remainder of the horizon from the resulting belief. We can bound the final term in Eq. S26 by further application of the non-expansive property to rewards collected during macro-action execution:

$$\left\| \sum_{i=0}^{l-1} \gamma^i \Big( \mathbb{E}_{s \sim b_*^{A_{1:i}}}[R(s,a_i)] - \mathbb{E}_{s \sim b^{A_{1:i}}}[R(s,a_i)] \Big) \right\| \tag{S27}$$

$$\leq \sum_{i=0}^{l-1} \gamma^i \left\| \mathbb{E}_{s \sim b_*^{A_{1:i}}}[R(s,a_i)] - \mathbb{E}_{s \sim b^{A_{1:i}}}[R(s,a_i)] \right\| \tag{S28}$$

$$\leq \sum_{i=0}^{l-1} \gamma^i \int_{\mathcal{S}} \left\| R(s,a_i)(b_*^{A_{1:i}}(s) - b^{A_{1:i}}(s)) \right\| ds \tag{S29}$$

$$\leq \sum_{i=0}^{l-1} \gamma^i R_{max} \int_{\mathcal{S}} \left\| (b_*^{A_{1:i}}(s) - b^{A_{1:i}}(s)) \right\| ds \tag{S30}$$

$$\leq \sum_{i=0}^{l-1} \gamma^i R_{max} \left\| b_*^{A_{1:i}} - b^{A_{1:i}} \right\|_1 \tag{S31}$$

$$\leq \sum_{i=0}^{l-1} \gamma^i R_{max} k^i \delta = \frac{1 - \gamma^l k^l}{1 - \gamma k} R_{max} \delta \tag{S32}$$

Plugging this expression in to Eq. S23, we have the recursion: $\eta_h \leq L\delta_{\mathcal{B}} + \frac{1 - \gamma^l k^l}{1 - \gamma k} R_{max} \delta_{\mathcal{B}} + \gamma^l Lk^l \delta_{\mathcal{B}} + \gamma^l \eta_{h-l}$. This expression depends on $l$, the length of the optimal macro-action at horizon $h$, in a a complex way. Because $l$ is variable and unknown *a priori*, we replace $l$ with its case value in each expression: $\eta_h \leq L\delta_{\mathcal{B}} + \frac{R_{max}\delta_{\mathcal{B}}}{1 - \gamma k} + \gamma Lk\delta_{\mathcal{B}} + \gamma \eta_{h-1}$. The result follows by expanding this recursion with $\eta_0 = 0$. □

## C  Macro-actions in Discrete Problems

The main text belief-dependent macro-actions for general POMDP problems by leveraging a representation of the optimal POMDP value function. However, for discrete POMDP problems (problems in which the state, action, and observation spaces are discrete), the value function is known to have special piecewise-linear and convex structure [1], and macro-actions can be generated using this PWLC representation of the value function. The analysis section can be adapted directly; in discrete problems the PWLC value function is Lipschitz continuous with respect to the 1-norm on discrete belief states for Lipschitz constant $L \leq \frac{1-\gamma^H}{1-\gamma} R_{\max}$.

In the following section, we construct an approximation to the optimal PWLC value function by iteratively backing up a set of $\alpha$-vectors over a finite horizon $H$. At each backup horizon $h$, the set of vectors $\hat{\Gamma}_h^*$ contains a mixed set of open-loop and closed-loop vectors.

### C.1  Value Iteration in Belief Space

Despite it's continuous nature, the value function for any discrete, finite horizon POMDP can be represented by a piecewise-linear and convex function (PWLC) with a finite number of supporting hyperplanes, often called $\alpha$-vectors. This property can be observed directly from Eq. 2 when integration is replaced by summation. For each control action $a$, construct an $\alpha$-vector: $\alpha_a = [R(s_1, a), \ldots, R(s_N, a)]^\top$ for each state $s_1, \ldots, s_N \in \mathcal{S}$. Then:

$$V_0^*(b) = \max_{a \in \mathcal{A}} \alpha_a^\top b. \tag{S33}$$

As the maximum of $|\mathcal{A}|$ linear segments, $V_0^*$ is PWLC and is represented by the supporting hyperplanes $\Gamma_0 = \{\alpha_a\}_{a=0}^{|\mathcal{A}|}$. Value-iteration over belief-space proceeds by building the horizon-$h$ value function from the set of $\alpha$-vectors at horizon-$(h-1)$. At each step of value iteration, the resulting value function remains PWLC [1] As is standard in point-based POMDP methods [5], we maintain only the subset of the possible $\alpha$-vectors that dominate at the exemplar beliefs in $\mathcal{B}$, keeping $\Gamma_h$ at a constant size.

For discrete problems, we represent the transition function $T$ by a set of transition matrices $\{\mathbb{T}_a\}_{a=0}^{|\mathcal{A}|}$, such that $\mathbb{T}_a[i, j] = P(S_{t+1} = i \mid S_t = j, a_t = a)$. The observation function $O$ will be represented by a set of observation matrices $\{\mathbb{O}_a\}_{a=0}^{|\mathcal{A}|}$, such that $\mathbb{O}_a[i, j] = P(Z_t = i \mid S_t = j, a_{t-1} = a)$.

### C.2  Open- and Closed-loop $\alpha$-vectors

As in the main text, we constructed a value function $\hat{V}_h^*$, which represents the value of beliefs when acting under a policy that selectively leverages open-loop actions. In this discrete setting, $\hat{V}_h^*$ will be represented by a finite set of $\alpha$-vectors, $\hat{\Gamma}_h^*$.

We initialize $\hat{V}_0^*$ with the set of $\alpha$-vectors, $\hat{\Gamma}_0^*$, as described in Eq. S33. The backup operator constructs the set $\hat{\Gamma}_h^*$ from the set $\hat{\Gamma}_{h-1}^*$ via the following operations.

**Closed-loop**  First, we construct the standard, closed-loop $\alpha$-vectors, which represent the value function under closed loop dynamics [1, 5]. First, the one-step reward vectors are constructed for $a \in \mathcal{A}$, which represents the immediate reward of an action $a$:

$$\Gamma_h^{a,*} = \alpha_a. \tag{S34}$$

Then, a set of projected $\alpha$-vectors is constructed, which capture the effect of the observation and transition dynamics on an input belief.

$$\Gamma_h^{a,z} = \{\gamma \alpha_{h-1}^\top \operatorname{diag}(\mathbb{O}_a[z, :]) \mathbb{T}_a \mid \alpha_{h-1} \in \hat{\Gamma}_{h-1}^*\} \tag{S35}$$

Next, for each belief $b$ in the belief set $\mathcal{B}$ the optimal $\alpha$-vector under an action $a$ for that belief is computed by summing over possible realized observations:

$$\Gamma_h^a = \{\Gamma_h^{a,*} + \sum_{z \in \mathcal{Z}} \operatorname*{arg\,max}_{\alpha \in \Gamma_h^{a,z}} \alpha^\top b \mid b \in \mathcal{B}\}. \tag{S36}$$

Finally, only the optimal action and its associated $\alpha$-vector for each belief is maintained:

$$\Gamma_h = \left\{ \arg\max_{\alpha \in \{\Gamma_h^a | a \in \mathcal{A}\}} \alpha^\top b \mid b \in \mathcal{B} \right\}. \tag{S37}$$

**Open-loop**  To determine the VoI from a specific belief state, we introduce *open-loop $\alpha$-vectors*, which represent the deterministic transition of belief due to system dynamics in the absence of observations. These open-loop (OLP) $\alpha$-vectors are constructed similarly to their closed-loop counterparts, where the open-loop transition dynamics are governed by only the transition matrix:

$$\Gamma_h^{a,*,OLP} = \{\gamma \alpha_{h-1}^\top \mathbb{T}_a \mid \alpha_{h-1} \in \hat{\Gamma}_{h-1}^*\} \tag{S38}$$

$$\Gamma_h^{a,OLP} = \left\{ \Gamma_h^{a,*} + \arg\max_{\alpha \in \Gamma_h^{a,*,OLP}} \alpha^\top b \mid b \in \mathcal{B} \right\} \tag{S39}$$

$$\Gamma_h^{OLP} = \left\{ \arg\max_{\alpha \in \{\Gamma_h^{a,OLP} | a \in \mathcal{A}\}} \alpha^\top b \mid b \in \mathcal{B} \right\}. \tag{S40}$$

## C.3   Value Iteration Backups

At each backup during value iteration, we add a mixture of open- and closed-loop $\alpha$ vectors to our current vector set $\hat{\Gamma}_h^*$. For each belief $b \in \mathcal{B}$, we compute the open- and closed-loop value and the value of information:

$$V_h^{OLP}(b) = \max_{\alpha \in \Gamma_h^{OLP}} \alpha^\top \cdot b \tag{S41}$$

$$V_h^{CLP}(b) = \max_{\alpha \in \Gamma_h} \alpha^\top \cdot b \tag{S42}$$

$$\text{VoI}_h(b) = V_h^{CLP}(b) - V_h^{OLP}(b) \tag{S43}$$

If VoI $\leq \tau$, we add the corresponding open-loop vector to the set $\hat{\Gamma}_h^*$ and add belief $b$ to the open-loop set; otherwise, we add the closed-loop vector. Thus, at horizon $h$, we represent the value function $\hat{V}_h^*$ using a mixture of open- and closed-loop $\alpha$-vectors, representing regions of belief space in which open-loop actions are near-optimal.

## C.4   A Note Algorithmic Complexity

Point-based POMDP algorithms maintain a set of $\alpha$-vectors of constant size; the main algorithmic cost is construction of the updated set of vectors using the methodology described in the previous section. Traditional point-based methods have complexity $\mathcal{O}(|\mathcal{S}|^2|\mathcal{A}||\mathcal{Z}||\Gamma_{h-1}|)$ to generate intermediate $\alpha$-vectors and complexity $\mathcal{O}(|\mathcal{S}||\mathcal{A}||\mathcal{Z}||\Gamma_{h-1}||\mathcal{B}|)$ to selected the maximizing $\alpha$-vector for each belief in $\mathcal{B}$ [5]. During value iteration for VoI macro-action generation, evaluating open-loop actions and computing VoI has cost equivalent to adding one additional observation to the observation space that provides no information about the current state. This leads to an algorithmic complexity of $\mathcal{O}(|\mathcal{S}|^2|\mathcal{A}|(|\mathcal{Z}|+1)|\Gamma_{h-1}|)$ to generate intermediate $\alpha$-vectors and $\mathcal{O}(|\mathcal{S}||\mathcal{A}|(|\mathcal{Z}|+1)|\Gamma_{h-1}||\mathcal{B}|)$ to evaluate VoI and select the open- and closed-loop sets. For large state spaces, the algorithmic complexity of macro-action generation is dominated by constructing the closed-loop $\alpha$-vectors during value iteration.

# D Experiments

## D.1 Experimental Setup and Parameters

The agent can move in each of the cardinal directions or stay in place ($|\mathcal{A}| = 5$). The agent observes the target location with discretized Gaussian noise (diagonal covariance, $\sigma^2 = 6.25$, $|\mathcal{Z}| = 100$) in Experiment 1 and discretized Gaussian noise with diagonal covariance varying linearly depending on the agent's location, from $\sigma = 0$ (perfect observation) in the bottom row of the world to $\sigma^2 = 25.0$ in the top row of the world in Experiment 2. We plan over a horizon of 75 iterations for Experiment 1 and 25 iterations for Experiment 2. We have discount factor $\gamma = 0.99$. In Experiment 1, the reward function penalizes the squared L2 distance between the target and the agent, and in Experiment 2, the agent receives a reward of $50.0$ if it is in the cell as the target when the target is in the zone of interest ($\{0, 1\} \times \{0, 1\}$); otherwise, the agent receives zero reward. The VoI threshhold $\tau$ is set to $\tau = 5$ in Experiment 1 and $\tau = 0.05$ in Experiment 2.

Policies are constructed using point-based value iteration [5] with a fixed-size belief set $|\mathcal{B}| = 285$ and executed in a set of $M = 500$ tracking experiments for evaluation. The belief set $\mathcal{B}$ is initialized to beliefs reachable under a QMDP policy [4] and three iterations of alternating value-iteration and belief set updates are performed, in which the value function estimate is used to update the set of reachable beliefs $\mathcal{B}$, which in turn is used to improve the value function estimate [5]. These values were determined by computational constraints; value function convergence during experimentation was not assessed.

Although PBVI formed the base approximation algorithm for these experiments, any approximation of the POMDP value function can form the base of macro-action construction. Further improvement may be seen if algorithms such as SARSOP [2] that explicitly attempt to approximate the optimally reachable belief space are used as the base approximation.

## D.2 Additional Visualizations

The following figures present additional visualizations of the random walk and boundary dynamic experiments presented in Table 1. Fig. S1 provides scatter plots comparing the performance of the VoI macro-action policy and the best closed-loop policy. Each point in the scatter plot represents a paired experiment with identical target dynamics. These plots highlight the variability of the best closed-loop planner and the reduction in the empirical value of $\delta_{\mathcal{B}}$ under the VoI macro-action policy. Fig. S1 additionally presents histograms showing the proportion of the planning horizon spent in open loop and the length of macro-actions taken by the agent for each dynamic under the VoI macro-action policy. These plots highlight the utility of using VoI to selectively act in open loop; the VoI macro-action policy acts in open-loop for a large fraction of the planning horizon, using extended sequences of open-loop actions, and yet performs at least as well as the best closed-loop planner.

*Figure S1: **Macro-actions in (A) boundary versus (B) random walk dynamics**: (Left) Performance for the VoI MA policy versus the Best CL policy in paired experiments. Points in the green region represent experiments for which VoI MA outperforms Best CL, achieving higher realized reward or lower values of $\delta_{\mathcal{B}}$; red regions represent experiments in which Best CL outperforms VoI MA. Color indicates point density (yellow high density to blue low density). (Right) Histograms of the proportion of the planning horizon the VoI MA planner spends in open loop and the length of all executed VoI macro-actions across $M = 500$ experiments.*