[Reviews · NeurIPS 2020]

Review 1

Summary and Contributions: This paper describes an approach whereby an agent uses open-loop planning to create macro-actions chaining sequences of actions together in order to reduce the complexity of offline planning in POMDPs. The novelty of the approach is in knowing when to create macro-actions -- where the value of information (VOI) earned by the sequence of actions is low. Importantly, theoretical analysis provides a bound on the regret incurred by planning such macro-level actions, where the bound is a function of key parameters of the open loop planning and VOI thresholds, enabling a designer to fine tune the approach, trading of reduced complexity with an acceptable error bound. Empirical results in a dynamic tracking domain highlight improvements in cumulative rewards earned by the agent when planning with the approach.

Strengths: The paper is supported by both theoretical and empirical analysis, and I especially appreciated the care with which the theoretical analysis was written -- it was easy to follow and well explained. The approach to determining when macro-actions can be generated using low VOI is novel, interesting, and a principled contribution to the combination of open and closed loop planning in POMDPs. The paper will be relevant to the single agent planning community at NeurIPS and could possibly inspire extensions to multiagent planning.

Weaknesses: I would have appreciated a more extensive empirical analysis to see how well the approach works in a wider variety of domains. In many classic benchmark POMDPs problems, VOI is an important guide to agent planning, e.g. in RockSample (Smith & Simmons, 2004) and especially UAV Navigation (Ong et al., 2009). I believe that both domains include opportunities for macro-actions of length between the lengths of the boundary and random walk dynamic experiments in the paper.

Correctness: The claims and methods are mostly correct. In Table 1, it is unclear whether a correction for multiple tests was employed when statistical significance was evaluated. Since more than two approaches were compared, a method such as the Bonferroni correction should be used when making pair-wise comparisons between the methods for each setting (within boundary dynamics, and within random walk dynamics). Otherwise, the true p from the combined Welsh t-tests will be higher than the p-value output by any single test (i.e., the risk of Type 1 error is increased).

Clarity: The paper is very well written and interesting to read.

Relation to Prior Work: The relevant related work is cited and discussed.

Reproducibility: Yes

Additional Feedback: My questions for the authors include: 1) Did you perform a correction such as Bonferroni in your statistical significance tests? 2) In your experimental setup, why did you use PBVI as the base planning algorithm, instead of one of the later point-based methods, especially SARSOP since it maintains a tighter reachable belief space (an important discussion in Section 1) 3) How did you decide on the hyperparameters for your experimental setup, e.g. to use a fixed number (285) belief points, and only updated the set of alpha vectors representing the value function estimate three times (was this enough for convergence)? ##### Post Rebuttal ##### I thank the authors for their clarifying responses to my questions! The answers will help strengthen an already good paper.


Review 2

Summary and Contributions: The paper provides an algorithm to generate macro-actions autonomously to reduce the expanded belief space for the POMDP solver and consequently reduce the complexity of it. The algorithm comes with a performance guarantee. Finally, the method is tested on a toy problem.

Strengths: The fact that macro-actions were generated automatically makes the method generalizable to many domains. Also, the method comes with a theoretical performance guarantee.

Weaknesses: The empirical section needs to be improved by including standard test-benchmarks in the POMDP domain, many of them available online. If standard test sets such as Rock_Sample, Tag, or others that are available in SARSOP website do not meet the criteria for VOI, the authors can try the 4 navigation tasks on MIT CSAIL floors (e.g. forth.pomdp in http://www.pomdp.org/examples/). Related to the test sets, it looks like VoI works when the observations are not that much informative and transitions are not very noisy. While this could be true in many practical applications, it is kind of contrary to the core concept of POMDPs. As a result, I think a venue specialized for planning or robotics (such as RSS) would be a more suitable option for this work.

Correctness: yes, but I think the results should be compared to SARSOP as well because SARSOP also tries to focus on the reachable belief state.

Clarity: Yes, the paper is clearly written.

Relation to Prior Work: yes

Reproducibility: Yes

Additional Feedback: It looks like the authors compared their method to the vanilla point-based method (2003 paper), but called it the best-closed loop. Did they use this because it was the V* used in their algorithm? What's the result of SARSOP? What is the result of VOI if SARSOP is used for V*? Based on authos' feedback and other reviews, I increased my score. I request the authors to change the misleading term of "best closed-loop" as they promised in their feedback.


Review 3

Summary and Contributions: This paper introduces a new method for computing action policies via discovering macro-actions in POMDP domains. More specifically, the concept of "value of information", which is different from the traditional concept from decision-making theory, was introduced to assist with discovering macro-actions. Intuitively, when the expected cumulative reward is low (i.e., VoI is low), macro-actions are formed. The developed approach was evaluated using a dynamic tracking problem. Results show that it performed better than "best closed-loop" and "fixed length macro-action" baselines. ------------ After rebuttal -------------- Based on the response letter and PC discussion, the overall score has been moved up.

Strengths: POMDP is a general decision-making framework, and improving the performance of POMDP systems can potentially produce good impact with significance. The developed algorithm (VoI MA) is intuitive, novel, and makes sense. The relevance to the NeurIPS community is good as planning and decision making has been one of the focuses of the conference.

Weaknesses: The work is not well presented. Terms like open-loop actions, closed-loop policies, and reachable belief space were used without definitions provided. As a result, the reviewer had difficulties understanding Figures 1 and 2. Value of information is the key of this work, but was only briefly discussed in Section 4.1. The major concern is on the evaluation of the developed methods. The POMDP community has provided a number of benchmark problems. It's unclear how the single evaluation domain of "dynamic tracking" was selected for experiments. The reviewer expected more benchmarks to be included in experiments, where the benchmarks can potentially be categorized based on their properties (horizon, reward, state size, etc). The experiment setup could be similar to those papers from the literature on general-purpose POMDP methods, such as those that introduced point-based methods, POMCP, and SARSOP. Also, the performance improvement was quite marginal in the "random walk" setting, which further raises questions on the effectiveness and applicability of the developed approach. Actually it looks like the dynamic tracking problem is not the best for demonstrating the effectiveness of the developed approach. Those domains that have been used in hierarchical RL, such as taxi and four-room, can be better for highlighting the VoI concept. In those domains, the navigation actions over cells can potentially form macro-actions due to their low cumulative rewards. It's unclear how this only testing domain was selected in this paper.

Correctness: The developed approach looks correct, though the usefulness was not clearly demonstrated through either demonstrations or statistical results.

Clarity: Not quite.

Relation to Prior Work: Yes.

Reproducibility: Yes

Additional Feedback:


Review 4

Summary and Contributions: This paper presents a point-based value iteration algorithm that leverages open-loop macro-actions at beliefs with low information-of-value (VOI) to achieve near-optimal performance. This is achieved by simultaneously perform open-loop and close-loop value updates and explicitly labeling sampled beliefs as open- or close-loop according to the VOI. The VOI of a belief is conveniently calculated as the gap between open-loop and close-loop values. "Open-loop actions have bounded regret exactly when VoI is low." This is simple and insightful. For most people with sufficient experience with POMDP planning, it is quite intuitive as well. However, turning it into an algorithm and, more importantly, an algorithm with theoretical performance guarantee, is a nice surprise.

Strengths: The idea of switching to macro-actions according to value-of-information is insightful and novel. The theoretical guarantee provided is also highly desirable.

Weaknesses: The experiments fall short, in comparison with the novel idea and the theoretical results. I would encourage the authors to devote some efforts to experimental evaluation (e.g., based on the three conditions listed in the paper) to bring out the full impact of this very nice idea.

Correctness: I am sure of the idea and its potential impact, even though the experimental part can use some additional work.

Clarity: Well motivated, clear.

Relation to Prior Work: Yes. Some additional ones: K. Hsiao, L.P. Kaelbling, and T. Lozano-P ́erez. Grasping POMDPs. In Proc. IEEE Int. Conf. on Robotics & Automation, 2007. Z.W. Lim, D. Hsu, and W.S. Lee. Monte Carlo value iteration with macro-actions. In Advances in Neural Information Processing Systems. 2011.

Reproducibility: Yes

Additional Feedback:

[Author Response · NeurIPS 2020]

We thank the reviewers for their insightful feedback. We are encouraged that they find the proposed algorithm for
generating value of information (VoI)-based macro-actions in POMDPs novel (*R1*, *R3*, *R4*), intuitive (*R3*, *R4*), and the
theoretical analysis carefully written and explained (*R1*). Reviewers highlighted the generalizability to many domains
(*R2*), including multiagent planning (*R1*), and the benefit to the community of the theoretical regret bound (*R1*, *R2*, *R4*).
We address reviewer comments below and begin by situating the paper's intended contribution:

• **What is our goal?** We introduce a metric (value of information (VoI)) for quantifying when the information provided
by sensing is most useful in partially observable planning problems. We utilize this metric to develop an algorithm
with accompanying theoretical performance guarantees for generating and effectively using open-loop macro-actions
in partially observable planning problems without sacrificing policy performance.

• **Why is this our goal?** VoI is highly variable within many planning problems – task performance relies crucially on
information-gathering from some belief states, and significantly less so from others. We show how to exploit this
property to reduce planning complexity in a way that current POMDP algorithms do not. With VoI macro-actions,
POMDP planners incur the complexity of full, closed-loop planning only when necessary. Counter to (*R2*), that low
VoI is "contrary to the core concept of POMDPs", VoI macro-actions expand the set of problems that can be efficiently
solved using POMDP methods, but are guaranteed to recover the performance of closed-loop POMDP planners in
classical domains such as Tiger that exhibit uniformly high VoI.

• **What is not our goal?** The primary critique of reviewers is the limited scope of our experimental results. (*R1*, *R2*, *R3*)
suggest validation on standard POMDP benchmark problems. While benchmark problems are certainly important, the
dynamic tracking problem allows us to directly vary the value of information and evaluate algorithm performance. The
goal of these experiments is not to outperform state-of-the-art baselines on standard POMDP tasks, but to demonstrate
that there is a class of problem for which reducing the size of the reachable belief space using VoI macro-actions can
be achieved without sacrificing policy performance.

We agree that further empirical validation of our work is an important next step. However, given limited space, we
chose to focus on providing a clear and extensive treatment of our algorithm and theoretical results, which reviewers
highlight (*R1*, *R2*, *R4*). The reviewers state that they are "sure of the idea and its potential impact" (*R4*) and that it
represents a "novel, interesting, and principled contribution" (*R1*), despite the limited experiments. All reviewers agree
on the novelty and impact of the algorithm and analysis, which we believe to be a significant contribution.

**1. Experimental Scope - Dynamic Tracking Experiments** (*R1*, *R2*, *R3*, *R4*) Unlike fixed POMDP benchmarks, the
dynamic tracking experiments allow us to evaluate planner performance across a spectrum of VoI (conditions (i)-(iii)
suggested in the text). The goal of these experiments is to validate the performance guarantees presented in Theorem
5.2 — not to compete with state-of-the-art POMDP algorithms on benchmark problems (many of which are designed to
test planner performance under uniformly high VOI conditions). We agree that exploring the VoI structure in a few of
the benchmark POMDP problems mentioned by reviewers (*R1*, *R2*, *R3*) is an interesting avenue for future work.

(*R3*) points to the marginal improvement over closed-loop in our random walk domain — however, random walk
was included precisely to demonstrate our theoretical claim that VoI macro-action polices recover the performance of
closed-loop POMDP algorithms in high VoI problems like the random-walk experiment. We will clarify this in the text.

(*R4*) was curious about experimental results under different VoI regimes – we are also excited about these results and
will include an ablation study in the camera-ready under the low VoI conditions introduced in the text.

**2. Experimental Scope - State-of-the-art Algorithms** (*R1*, *R2*, *R3*) Reviewers request a comparison to state-of-
the-art POMDP baselines, such as SARSOP. We emphasize that our algorithm and theoretical results hold for any
approximation of the closed-loop value function, including that produced by SARSOP. Currently, we use PBVI as the
base closed-loop algorithm for macro-action construction - therefore, comparing to PBVI directly demonstrates of the
benefit of open macro-actions. We will modify the potentially misleading terminology "best closed-loop".

(*R1*, *R2*) thoughtfully suggest using SARSOP as the base algorithm for generating macro-actions. This is an interesting
idea — SARSOP is designed to more effectively *search* a policy's reachable belief space; VoI macro-actions actually
*shrink* this reachable belief space. Combining these complimentary strategies is an exciting area of future work.

**3. Multiple-hypothesis testing** (*R1*) We have updated the significance test to control the false positive rate of both
comparisons using the Bonferroni correction — the experimental claims and significance comparison remain the same.

**4. Hyperparameters** (*R1*) Hyperparameter settings were dictated by computational constraints in this larger experi-
mental domain. All algorithms are run with the same hyperparameters to evaluate the impact of VoI macro-actions.

**5. Definition of terms** (*R3*) Definitions of common terms, such as open- and closed-loop, are omitted due to space
limitations. For definition of optimally reachable belief space (RBS*) and VoI, see Ln. 19 and Eq. (4) respectively. We
agree that the figure captions should be more self-contained, and will add a definition for RBS* in the caption of Fig. 1.

[Meta-Review · NeurIPS 2020]

The authors did a good jump of addressing reviewer concerns in the response. There were some lingering concerns about whether the authors had picked the best compare-to choices for their experiments. Additional experiments and/or more careful justification for the choices made would always help. I would recommend that the authors take the reviewers' comments into account in preparing the final version of the paper.